# Dual-Wavelength Polarization-Dependent Bifocal Metalens for Achromatic Optical Imaging Based on Holographic Principle

**DOI:** 10.3390/s22051889

**Published:** 2022-02-28

**Authors:** Jiaqi Qu, Huaijian Luo, Changyuan Yu

**Affiliations:** 1The Photonics Research Center, Department of Electronic and Information Engineering, The Hong Kong Polytechnic University, Hong Kong 999077, China; jiaqi.qu@connect.polyu.hk (J.Q.); huaijian.luo@connect.polyu.hk (H.L.); 2The Hong Kong Polytechnic University Shen Zhen Research Institute, Shenzhen 518057, China

**Keywords:** dual-wavelength focusing, bifocal metalens, achromatic optical imaging, holographic principle, polarization-dependent

## Abstract

Recently, ultrathin metalenses have attracted dramatically growing interest in optical imaging systems due to the flexible control of light at the nanoscale. In this paper, we propose a dual-wavelength achromatic metalens that will generate one or two foci according to the polarization of the incident. Based on geometric phase modulation, two unit cells are attentively selected for efficient operation at distinct wavelengths. By patterning them to two divided sections of the metalens structure plane, the dual-wavelength achromatic focusing effect with the same focal length is realized. In addition, the holographic concept is adopted for polarization-dependent bifocal generation, in which the objective wave is originated from two foci that are respectively formed by two orthogonal polarization states of circularly polarized light, namely Left-handed circularly polarized (LCP) light and Right-handed circularly polarized (RCP) light. The incident light is considered as the reference light. The achromatic focusing and polarization-dependent bifocusing are numerically verified through simulations. The proposed design opens the path for the combination of multi-wavelength imaging and chiral imaging, which may find potential applications, such as achromatic optical devices and polarization-controlled biomedical molecular imaging systems.

## 1. Introduction

Metasurface has emerged as indispensable optical elements in recent years with its ultrathin structures and flexible capability in light manipulation. Composed of subwavelength nanostructures, optical metasurfaces feature the exotic capability to arbitrarily modulate the light properties, such as the phase, amplitude and polarization at the nanoscale [1,2,3,4,5]. Due to their superiority in wavefront modulation, a plethora of planer optical devices have been extensively investigated, including metahologram [6,7,8], beam steering [9], vortex generators and detectors [10,11,12] and polarization analysis [13].

In particular, research endeavors have been devoted to the investigation of metalens design for its significant role in the optical imaging field. Versatile designs on metalenses have been reported in the visible [14,15,16] and near-infrared wavelengths [17,18] with plenty of applications and integrations. In our previous work, we demonstrated the dielectric metalens integrated with optical fiber working in the near-infrared range [19,20].

However, the above investigations are mainly focused on single focus design. Recently, there have been growing numbers of explorations on multi-focal metalens, aiming to enhance the versatility of the metalens design strategy. With generated foci at different transverse/longitudinal positions, a multi-focal metalens is able to provide flexibility and diversity in practical applications, such as multi-imaging system [21,22] and chiral imaging [23,24].

Section division with different foci is one of the most commonly used strategies [25,26,27]. In this case, the generated foci are fixed and cannot be controlled by the incident. To overcome the aforementioned problem, the method of combining propagation phase and geometric phase to simultaneously tailor the phase distribution is proposed [28,29]. The two foci formed by Left-handed circularly polarized (LCP) light and Right-handed circularly polarized (RCP) light are polarization-dependent according to the incident chirality. Subsequently, to simplify the design and fabrication, a transverse bifocal metalens is introduced by Zhou [30] based on geometric phase and the holographic principle. Using a single unit cell with different rotation angles, the metalens efficiently realizes the polarization-dependent focusing, making the bifocal metalens design simple and practical.

The aforementioned bifocal metalenses only work at a single wavelength. Due to the chromatic aberration originated from their diffractive nature [31,32], meta-devices heavily depend on the designed operation wavelength. The practical application of metalenses in multispectral optical devices may be greatly constrained. There are plenty of methods to achieve achromatic central focusing, such as multi-layer metalenses [33] and doublet metalenses [34,35]. For example, in Acayu’s work [36], three layers with different design parameters are stacked to achieve optimal interaction with multi-wavelength. In addition, spatial interleaving [37] and segment [21] were also introduced to satisfy different phase profiles for multi-wavelength. However, there have been few attempts at combining bifocusing and achromatic focusing via single-layer metalenses.

In this paper, we adopt the spatial segment method for dual-wavelength achromatic focusing, combined with the holographic principle and geometric phase to generate polarization-dependent bifocal spots. As far as we know, this is one of the first investigations on realizing bifocus generation and achromatic focusing simultaneously. According to the transmission and polarization conversion efficiency under the illumination of circularly polarized light, we attentively choose two unit cells working at discrete wavelengths. The selected two elements, made of Silicon nanobricks on the glass substrate, are arranged into divided sections to accomplish a dual-wavelength achromatic focusing effect.

The holographic principle is used to motivate polarization-dependent bifocal spots generation. The foci formed by RCP and LCP can be controlled by the polarization of the incident, which is more flexible in practical utilizations. Our design combines the dual-wavelength achromatic focusing and polarization-dependent bifocal design with the single-layer metalens, and this design is verified by the simulation results. We strongly believe that the proposed strategy will pave the way for the potential applications in multi-wavelength achromatic imaging systems, biomedical chiral imaging devices and virtual reality (VR)/augmented reality (AR).

## 2. Designs and Structures

The schematic illustration of the designed dual-wavelength achromatic polarization dependent bifocal metalens is shown in Figure 1a. The incident with wavelengths of 1000 and 1550 nm can be both focused into two focal spots with the same focal length. The unit cell utilized here is a Silicon nanobrick patterned on the glass substrate, which is shown in Figure 1b. There are basically two selected unit cells with different lattice constant (P), length (L) and width (W) in this design. Each unit cell provides a high operation efficiency at one specific wavelength while providing a low efficiency at another wavelength.

By spatially distributing the patterned area to different unit cells, a dual-wavelength achromatic metalens can be realized. As shown in Figure 1c, the metalens is divided into two parts for two different unit cells: one is the circle and the other is the remaining part of the square that removes the central circle. The radius of the circle is set to 7 µm and the side width of the square is 20 µm. In each section, the pattered unit cells rotate with different angles based on the geometric phase and holographic principle to enable polarization-dependent bifocal generation.

To impart the desired phase profile, geometric phase is introduced by rotating the nanobricks with different angles. The principle can be derived from the general Jones matrix,
(1)$$J=R\left(\alpha \right)\left[\begin{array}{cc} t_{u} & 0\\ 0 & t_{v} \end{array}\right]\;R\left(-\alpha \right)=\left[\begin{array}{cc} \cos\alpha & -\sin\alpha \\ \sin\alpha & \cos\alpha \end{array}\right]\left[\begin{array}{cc} t_{u} & 0\\ 0 & t_{v} \end{array}\right]\;\left[\begin{array}{cc} \cos\alpha & \sin\alpha \\ -\sin\alpha & \cos\alpha \end{array}\right],$$
where α is the orientation angle of the unit cell with respect to the original axis. $t_{u}$ and $t_{v}$ are the transmission coefficients along the fast axis and slow axis of the unit cell. R(α) and R(−α) express the rotation matrix, which consists of Trigonometric functions of α.
(2)$$E_{T}^{R/L}=J*E_{I}^{R/L}=\frac{t_{u}+t_{v}}{2}E_{I}^{R/L}+\frac{t_{u}-t_{v}}{2}e^{\mp 2\alpha }E_{I}^{L/R}$$

If the incident light is LCP or RCP, the output field of the transmitted light is shown in Equation (2). The conversion of a different polarization will introduce a phase shift of 2α when the unit cell rotates α, while the transmitted light with the same polarization of the incident will experience no phase shift. By tailoring the rotation angle of the unit cell, a 2π phase span can be realized by the converted light. In this case, the unit cell will serve as a half-wave plate to ensure a high polarization conversion efficiency (PCE).

We carefully selected two different unit cells. The optimized parameters of unit cell 1 are $L_{1}=250\;\mathrm{nm}$, $W_{1}=100\;\mathrm{nm}$ and $P_{1}=400\;\mathrm{nm}$. As for unit cell 2, the parameters are $L_{2}=500\;\mathrm{nm}$, $W_{2}=150\;\mathrm{nm}$ and $P_{2}=600\;\mathrm{nm}$. The height of the two unit cells is set to 800 nm, which ensures a convenient platform for further fabrication. The transmission and PCE of two unit cells covering the wavelength range from 990 to 1600 nm are shown in Figure 2. The incident light is set as RCP, and thus the PCE stands for the ratio of the intensity of the converted LCP to the incident intensity.

Transmission describes the sum of the intensity of the converted light and the remaining light to the incident. Since the geometric phase works based on polarization conversion, it is clear that less difference between the transmission and PCE at a specific wavelength introduces a high operation efficiency. In Figure 2, green stars highlight the transmission and PCE of the unit cell at the wavelength of 1000 nm, and black ones highlight the same data at the wavelength of 1550 nm. It is worth noting that the unit cells are selected elaborately to exhibit a high operation efficiency at one wavelength while having a low operation efficiency at another wavelength.

To be specific, unit cell 1 (shown in Figure 2a) exhibits a small difference between the transmission and PCE at 1000 nm, indicating that a large amount of incident (around 90%) has been converted into the orthogonal polarization. Thus, high operation efficiency is realized at the wavelength of 1000 nm. Conversely, unit cell 1 shows a low operation efficiency at the wavelength of 1550 nm. It is derived from Figure 2b that unit cell 2 operates at 1000 nm with a low efficiency while working well at 1550 nm.

## 3. Results and Discussions

For a general central-focused metalens, the phase profile follows,
(3)$$\varphi \left(x,y\right)=-\frac{2\pi }{\lambda }\left(\sqrt{x^{2}+y^{2}+f^{2}}-f\right)$$
where f is the focal length of the designed metalens, (x,y) are the coordinates of unit cells in the whole structure, and λ is the operation wavelength. Here, a metalens working at 1000 nm uses unit cell 1 and a metalens working at 1550 nm uses unit cell 2. The designed focal length is 12 µm. The required phase profile of two lenses along x-axis at y is 0 are illustrated in Figure 3a. Based on the two phase distributions, unit cells are spatially rotated with different angles of φ(x,y)/2. The simulated intensity distributions of the x–z plane at y equals to 0 of two lenses are also shown in Figure 3b,c. It is clear from the tightly focused spots that both unit cells operate well in single focus metalens design at their specific wavelengths.

### 3.1. Dual-Wavelength Achromatic Metalens

It is generally known that, for a specific phase distribution, incidents with different wavelengths will focus with different focal lengths, which is clearly described in the following equation,
(4)$$\varphi \left(x,y\right)=-\frac{2\pi }{\lambda_{1}}\left(\sqrt{x^{2}+y^{2}+f_{1}{}^{2}}-f_{1}\right)=-\frac{2\pi }{\lambda_{2}}\left(\sqrt{x^{2}+y^{2}+f_{2}{}^{2}}-f_{2}\right)$$
where $\lambda_{1}$ and $\lambda_{2}$ are different operating wavelengths, and $f_{1}$ and $f_{2}$ are their corresponding focal lengths. To establish a dual-wavelength achromatic metalens, we spatially divide the metalens into two areas working at two individual wavelengths. It is clear in Figure 1c that the central circle will be allocated for unit cell 2, and the remaining part of the square will be allocated for unit cell 1. As discussed before, unit cell 1 works efficiently at 1000 nm but does not work well at 1550 nm. Conversely, unit cell 2 exhibits the opposite performance. With the elaborate spatial division, different phase profiles can be attained with a single metalens at different wavelengths with the same focal length, which is the critical issue for the dual-wavelength achromatic focusing.

Following Equation (4), we assign two different phase profiles to the distinct areas. The allocation rule is the same as Figure 1c, with unit cell 2 pattered in the central center and unit cell 1 patterned in the remaining part of the square. The side width of the square is 20 µm, and the radius of the central circle is 7 µm. The focal length is set to 12 µm. According to the different required phase profiles, unit cell 1 and unit cell 2 are carefully arranged in the two areas. To verify the achromatic focusing, RCP light at 1000 and 1550 nm are launched successively as the incident.

The optical intensity at the x–z plane at y equals 0 is shown in Figure 4 with the wavelengths of 1000 nm (a) and 1550 nm (b), indicating a tightly focused spot at the desired focal length. The simulated focal lengths are 12.1 and 11.8 µm at the illumination of 1000 and 1550 nm. The normalized optical intensity profiles along *x*-axis are also extracted and shown in Figure 4c,d. The full width at half maximum (FWHM) can be obtained accordingly, which are 0.70 and 1.55 µm for two distinct wavelengths. To further investigate the performance, the focusing efficiency will be measured, which is defined as the ratio of light passing through the rectangular aperture with a side width of three times of FWHM of the focal spot.

For incidents at 1000 and 1550 nm, the focusing efficiency are 40% and 27%, respectively. The difference of efficiency results from the unequal proportion of area allocated for unit cell 1 and unit cell 2. It is worth noting that the shape of the focal spot is slightly different from that of the normal focal spot (see Figure 3). Although we carefully selected the unit cells to make sure they work at distinct wavelengths, the operation efficiency did not fall to zero for both unit cells. There still exists a small amount of light at 1550 nm transmitting through unit cell 2 and obtaining the phase shift, and vice versa. In this case, the focusing effect will experience slight deviations, bringing about small changes in the shape of the focal spot.

### 3.2. Polarization-Dependent Bifocal Metalens Using the Holographic Principle

To simultaneously realize the bi-focusing, the holographic principle is utilized. The light from two focal spots can be considered as the object lightwave, while the incident plane wave serves as the reference light. The interfered light distribution, including phase and amplitude, which is commonly known as the hologram, can be derived subsequently. For simplicity, the amplitude of the interference light is regarded as a constant. The method is proposed and proved in [38]. The field distribution of the designed metalens is attained through the interference between the lightwave generated of the two foci and the incident. Here, the incident is the plane wave, and the generated two foci essentially correspond to RCP and LCP light. The phase distributions of the two foci are expressed as below:(5)$$\varphi_{LCP}\left(x,y\right)=-\frac{2\pi }{\lambda }\left(\sqrt{x^{2}+y^{2}+f_{1}{}^{2}}-f_{1}\right)$$
(6)$$\varphi_{RCP}\left(x,y\right)=-\frac{2\pi }{\lambda }\left(\sqrt{x^{2}+y^{2}+f_{2}{}^{2}}-f_{2}\right)$$
where $f_{1}$ and $f_{2}$ are the focal length of the generated foci. The proposed design is not only a simple bifocal lens but also exhibits a polarization-dependent characteristic. Two transverse focal spots will occur with the incident of the linear polarized light. The circularly polarized light will bring about the left focus (formed by LCP) or the right focus (formed by RCP). We set the phase profile of RCP with an additional phase shift of π. In this case, the phase modulation of the proposed design can be described as,
(7)$$\varphi \left(x,y\right)=angle(\exp(\varphi_{LCP}\left(x,y\right)+\exp\left(-\varphi_{RCP}\left(x,y\right)\right)$$
where the angle is the function of extracting the phase. The same method was proposed by Zhou [30] in their bifocal design. Following the above phase distribution, two foci at different positions with the opposite polarization states will be generated.

We first design the metalens for longitudinal bifocal spots generation. Compared to the transverse bifocal design, the longitudinal design experience only one variable, namely the focal length. For simplicity, we take the longitudinal design as the example to show the polarization-dependent focusing effect. Here, we utilize unit cell 2 and set the focal lengths of two foci to 8 and 13 µm. The designed metalens cover a square with the side length of 20 µm. The working wavelength here is set to 1550 nm as an example to illustrate the longitunal bifocal design.

Under the illumination of RCP and LCP, the transmitted light focused exactly at the designed two focal lengths, which are shown in Figure 5a,c. When the linearly polarized (LP) light serves as the incident, bifocal spots are obtained along the *z*-axis at 8.0 and 13.1 µm, manifesting a polarization-dependent characteristic. With the same processing method, a transverse bifocal design can also be implemented. We will explain the detailed performance of transverse bifocal design in the next chapter.

### 3.3. Performance of the Proposed Metalens

Based on the above discussions, we utilize the elaborately selected two unit cells for dual-wavelength achromatic focusing and use the holographic principle for the polarization-dependent bifocusing. Integrating the functionalities, the proposed dual-wavelength metalens for polarization-dependent bifocusing can be realized. In this case, we designed a transverse achromatic bifocal metalens with the focal length set to 12 µm. The distance between the generated foci and the center point at the focal plane, which is d in Figure 1a, is set to 5 µm. The size of the metalens and the allocation rules are shown in Figure 1c. The performances of the proposed design are shown in Figure 6 and Figure 7.

From Figure 6 and Figure 7, it is clear that the foci can be controlled by the incident polarization. When the incident is circularly polarized light, the generated focus appears at the left/right side of the focal plane, representing the chirality of the incident of LCP and RCP, respectively. While two foci will occur when the incident becomes linearly polarized light. Thus, the polarization-dependent bifocusing characteristic can be verified. The performance of the dual-wavelength achromatic focusing is further discussed in Figure 8.

For incident at 1000 nm, the focal length is 12.3 µm, and the distance between the generated foci and the center is 4.8 µm. As for illumination at 1550 nm, the focal length is 11.9 µm, and the distance between the generated foci and the center is 5.0 µm. With slight deviations, the coordinates of the generated foci coincide well the the initial design at two distinct wavelengths. Through numerical simulation, we collected the focusing efficiency of the two foci under different illuminations. The efficiencies of the left foci are 19% and 15% when illuminated under LCP at 1000 and 1550 nm, respectively. Similarly, the efficiencies of the right foci are 19% and 15% under RCP at 1000 and 1550 nm, respectively.

As for LP light, the left and right foci shares the same efficiency, which are 11% at 1000 nm and 8% at 1550 nm. Thus, the dual-wavelength achromatic focusing and polarization-dependent bifocal generation are attained simultaneously. To note, the focal spots become slender, which is mainly due to the non-zero operation efficiency of each unit cells at the wavelength they are not designed to work at. The slight deviations will not affect the functionalities of the proposed design. In general, the simulation results of the achromatic bifocal metalens coincide well with the desired strategy.

## 4. Conclusions

To conclude, we numerically proposed a dual-wavelength achromatic metalens that is able to generate polarization-dependent bifocal spots. Based on the geometric phase modulation, two elaborately selected unit cells were arranged in distinct areas to enable the achromatic focusing at two discrete wavelengths (1000 and 1550 nm) with the same focal length. The holographic principle was adopted to generate polarization-dependent transverse bifocal spots. We envision that this proposed strategy will find potential applications in biomedical chiral imaging systems, multi-wavelength achromatic devices, muti-foci imaging systems and VR/AR.

## Figures and Tables

**Figure 1 sensors-22-01889-f001:**
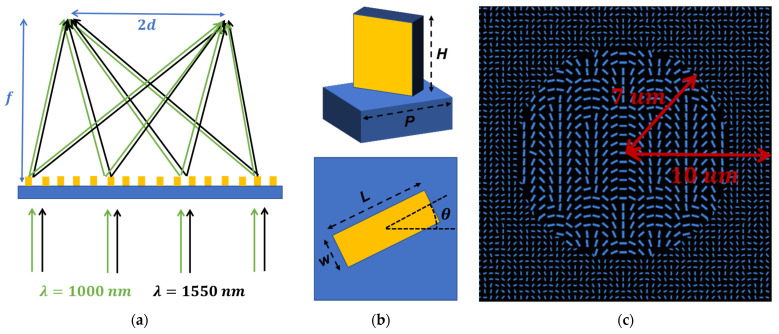
(**a**) Schematic illustration of the designed dual-wavelength achromatic polarization-dependent bifocal metalens. Transmitted light with the wavelengths of 1000 and 1550 nm can be both focused into the same focal plane transversely. (**b**) Unit cell with a Silicon nanobrick patterned on the glass substrate. (**c**) Spatial distribution of two different unit cells.

**Figure 2 sensors-22-01889-f002:**
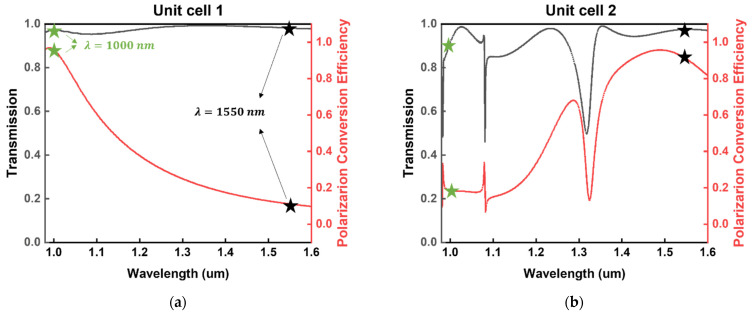
The transmission and PCE of the unit cell 1 (**a**) and unit cell 2 (**b**) with wavelengths ranging from 0.98 µm to 1.6 µm. Transmission and PCE are highlighted with green stars for 1000 nm and black stars for 1550 nm, respectively.

**Figure 3 sensors-22-01889-f003:**
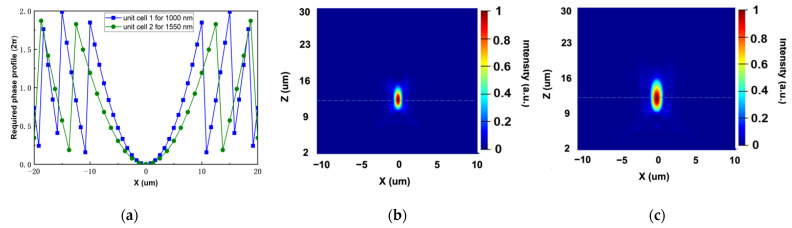
(**a**) The required phase profiles for the general single focus metalens under the illumination of 1000 and 1550 nm. (**b**,**c**) The normalized optical intensity of the x–z plane at y=0 with the incident at the wavelength of 1000 nm (**b**) and 1550 nm (**c**).

**Figure 4 sensors-22-01889-f004:**
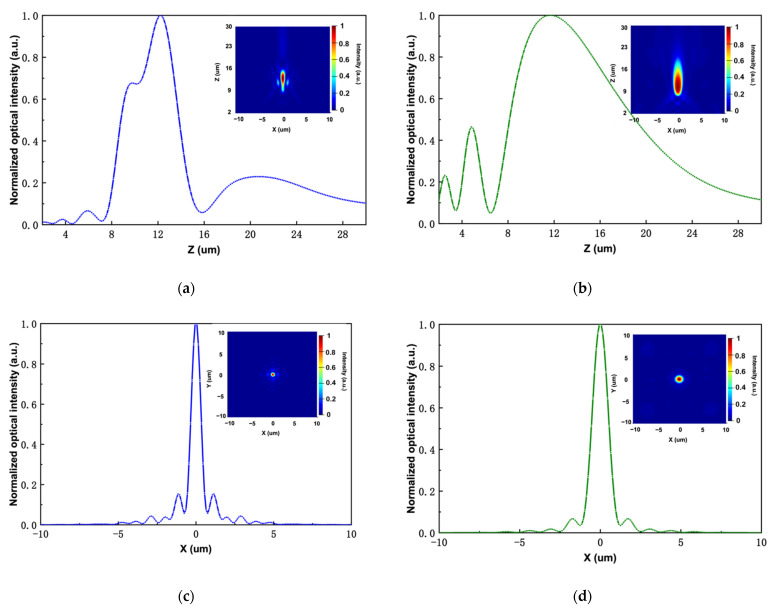
Performance of the dual-wavelength achromatic metalens. (**a**,**b**) The normalized optical intensity along *z*-axis at x=0 and y=0 with the illumination at 1000 and 1550 nm, respectively. The inset figures represent the normalized optical intensity along the cross section at the x-z plane when y=0. (**c**,**d**) The normalized optical intensity along *x*-axis at the focal plane. The inset figures illustrate the optical intensity at the focal plane.

**Figure 5 sensors-22-01889-f005:**
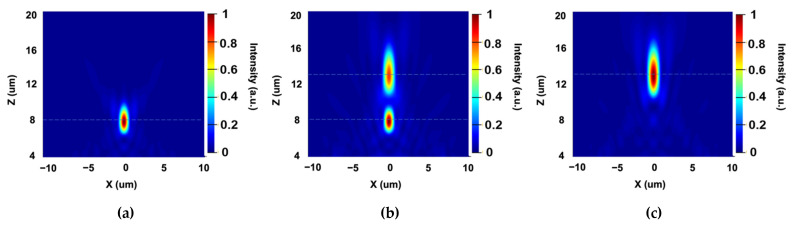
Simulated performance of the polarization-dependent bifocal metalens. (**a**–**c**) The intensity profile of the transmitted light field in the x–z plane when RCP, LP and LCP are incident.

**Figure 6 sensors-22-01889-f006:**
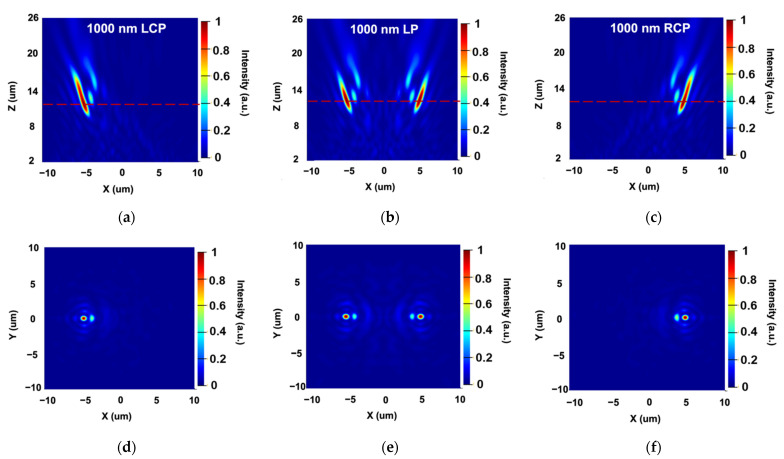
Performance of the dual-wavelength achromatic bifocal metalens at 1000 nm. (**a**–**c**) Simulated optical intensity of the proposed metalens with incident wavelength at the x–z plane when y is 0 for LCP, LP and RCP. (**d**–**f**) Simulated optical intensity at the x–y plane at the designed focal plane.

**Figure 7 sensors-22-01889-f007:**
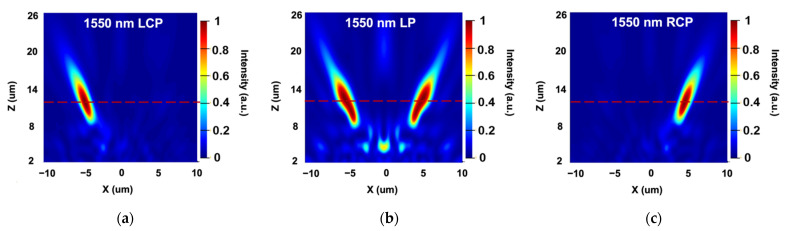
Performance of the dual-wavelength achromatic bifocal metalens at 1550 nm. (**a**–**c**) Simulated optical intensity of the proposed metalens with incident wavelength at the x–z plane when y equals 0 for LCP, LP and RCP. (**d**–**f**) Simulated optical intensity at the x–y plane at the designed focal plane.

**Figure 8 sensors-22-01889-f008:**
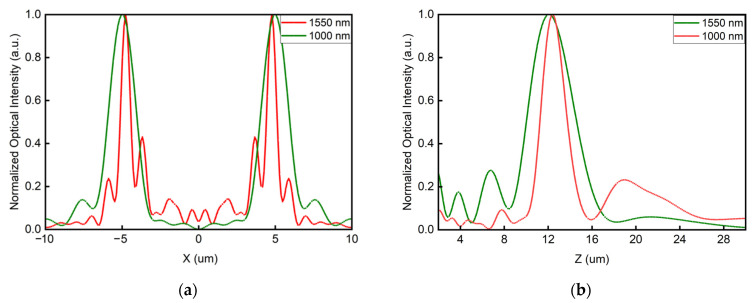
Optical intensity distribution under LP light. (**a**) Normalized optical intensity along *x*-axis at the focal plane. Red and green lines stand for different distribution at 1550 and 1000 nm, respectively. (**b**) Normalized optical intensity along z-direction when x is −5 µm and y is 0. Red and green lines stand for different distribution at 1550 and 1000 nm, respectively.

## Data Availability

The data that support the findings of this study are available from the authors on reasonable request.

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
