# Peer review of "Dual-Wavelength Polarization-Dependent Bifocal Metalens for Achromatic Optical Imaging Based on Holographic Principle"

_sensors, 2022, doi:10.3390/s22051889_

Round 1
Reviewer 1 Report
The authors show a dual-wavelength achromatic metalens that combines holographic principle and geometric phase to produce polarization-dependent bifocal spots. The writing level of the manuscript is commendable, but the current version has some negligible problems that must be explained and corrected. Therefore, I recommend reject in the present version.
- The introduction introduces several methods to realize the multi-wavelength achromatic metalens, and then shows the main content of the manuscript. But it is puzzling that numerous numerical simulations and experiments have demonstrated achromatic metalenses, and the authors do not focused on the difference of the manuscript from existing studies. In addition, the principles used in the manuscript are also very mature. Therefore, it is necessary to highlight the innovation of this paper in the introduction section.
- Specific parameters are indispensable for the designed metalens, while only the focal length is given, and the parameters such as the diameter of metalens are not listed. In addition, the two wavelengths chosen are 1000 nm and 1500 nm, what is the reason for the selection of specific wavelengths? In part 3.3, the focal points are produced on the left and right sides of the focal plane under the different incident lights, but the coordinates of the preset positions and the actual focal point positions are not given, and how much is the error of both.
- Figures in the manuscript also have some problems. For example, the incomplete display of intensity (a.u.) due to typography in Figures 6 and 7.
- Although the manuscript studies a dual-wavelength achromatic metalens, some characterization is missing. The focusing efficiency is very important for metalens, but it is not given in the manuscript.
Reviewer 2 Report
The manuscript describes the design and simulations of a metasurface for separating left and right-handed circular polarised light to different focal spots. This metasurface has been tested in simulations for two different wavelengths 1000 nm and 1550 nm.
The abstract and introduction need some careful polishing of the English language. The authors should make it clear that this is a design and simulation paper only, and they do not make an experimental demonstration of the device.
Line 91-92 and lines 159-160 please review the description of the unit cell. I do not understand the description of the "left area" as the unit cell seems to be the whole square?
I would also like to understand how the performance of the design would depend on the bandwidth of the incident light, or on the fabrication tolerance of these devices?
It is difficult to assess the efficiency of the device, because the simulations show normalised intensities. Can the authors comment on the overall efficiency of the device?
Can the authors discuss the contrast ratio, such as the side lobes in figures 4 and 6?
I recommend that the original design that is simulated should be made available as research data for this paper (eg in a supplementary section). I suggest that the image in Fig 1c should be adapted so that it represents the design, as it does not appear to be consistent with the parameters given in lines 114- 115. (Please check).
Round 2
Reviewer 1 Report
According to the reviewer's suggestion, the author carefully revised it one by one and suggested that the paper be accepted.